# Understanding the Limits of Vision Language Models Through the Lens of the Binding Problem

**Declan Campbell**[1], **Sunayana Rane**[2], **Tyler Giallanza**[2], **Nicolò De Sabbata**[3], **Kia Ghods**[1], **Amogh Joshi**[1], **Alexander Ku**[2], **Steven M. Frankland**[4], **Thomas L. Griffiths**[2,5*], **Jonathan D. Cohen**[1,2*], **and Taylor Webb**[6*]

[1]Princeton Neuroscience Institute
[2]Department of Psychology, Princeton University
[3]Department of Computer Science, EPFL
[4]Department of Cognitive Science, Dartmouth College
[5]Department of Computer Science, Princeton University
[6]Microsoft Research
* Equal contribution

## Abstract

Recent work has documented striking heterogeneity in the performance of state-of-the-art vision language models (VLMs), including both multimodal language models and text-to-image models. These models are able to describe and generate a diverse array of complex, naturalistic images, yet they exhibit surprising failures on basic multi-object reasoning tasks – such as counting, localization, and simple forms of visual analogy – that humans perform with near perfect accuracy. To better understand this puzzling pattern of successes and failures, we turn to theoretical accounts of the *binding problem* in cognitive science and neuroscience, a fundamental problem that arises when a shared set of representational resources must be used to represent distinct entities (e.g., to represent multiple objects in an image), necessitating the use of serial processing to avoid interference. We find that many of the puzzling failures of state-of-the-art VLMs can be explained as arising due to the binding problem, and that these failure modes are strikingly similar to the limitations exhibited by rapid, feedforward processing in the human brain.

## 1 Introduction

Recent progress in training large-scale neural networks on internet-scale datasets has led to the creation of AI systems with capabilities rivaling human performance across a broad range of complex tasks. Most recently, this has given rise to an array of vision language models (VLMs), including multimodal language models such as GPT-4v that can generate text descriptions of multimodal text and image inputs [1], and text-to-image models such as DALL-E 3 that can generate images from natural language descriptions [24]. However, despite the considerable success of VLMs across many tasks, these models still perform poorly on several surprisingly simple multi-object reasoning tasks – such as counting [23, 25, 40], relational image generation [7], relational scene understanding [15, 31], and simple visual analogy tasks [20, 38] – on which humans achieve near perfect accuracy.

Drawing from theoretical work both in cognitive science and neuroscience, we turn to the *binding problem* [9, 10, 29, 33, 36] as a potential explanation for these limitations. 'Binding' refers to the ability to associate one feature of an object (e.g., its color) with the other features of that object (e.g., its shape and location), and the 'binding problem' refers to the question of how the brain accomplishes this without interference between the features for different objects. It is widely recognized that the

human visual system relies on serial processing to solve this problem, iteratively directing attention to individual objects so as to avoid interference [28, 33], and that binding errors arise when it is forced to rely on rapid, parallel visual processing [14, 19, 33]. For example, when human participants are not able to effectively deploy serial processing (e.g., because attention is overloaded, or because speeded judgments are required), they are susceptible to so-called *illusory conjunctions* (e.g., mistakenly identifying a red square in an image that contains a green square and a red circle) [32].

In this work, we test the hypothesis that the failures exhibited by VLMs on multi-object reasoning tasks are due to representational interference resulting from an inability to manage the binding problem. We first investigate two classic tasks from the cognitive science literature, visual search [33] and numerical estimation [14, 19] (i.e., counting), finding that a wide range of VLMs (including 5 multimodal language models and 4 text-to-image models) exhibit stark capacity constraints similar to those displayed by human observers when forced to make speeded responses. Importantly, although these effects are more pronounced for scenes with more objects, they cannot be explained entirely as a function of the number of objects in a scene. Instead, we find that performance is best explained by the probability of interference given the specific distribution of features and their conjunctions within a scene. Motivated by this observation, we develop a novel scene description benchmark that systematically varies the likelihood of interference, finding that this quantity is highly predictive of binding errors.

We also apply these insights to better understand the limitations of VLMs in visual analogy tasks, introducing a simple input pre-processing technique to reduce the potential for representational interference. We show that this technique improves the performance of VLMs on the task, suggesting that their original limitation on this task may be due to a more basic difficulty with processing multi-object scenes, rather than an inability to process relations. Finally, we discuss the normative factors that underlie the binding problem [2, 9, 22], highlighting the role of compositional representations, which are useful for generalization, but introduce the potential for interference when shared representations are used to process multiple objects at the same time. We argue that, surprisingly, the binding failures exhibited by VLMs imply the presence of compositional representations. Overall, these results highlight the usefulness of cognitive science in helping to understand the limits of large-scale generative models, and suggest the presence of a common set of principles that govern information processing in both artificial systems and human cognition.

## 2   Visual Search

Extensive prior work in cognitive psychology has tested how people process scenes involving multiple objects and under what conditions their performance degrades. These studies demonstrate that performance is not driven solely by the number of objects present in a scene, but also depends on the likelihood of interference among objects given the specific distribution of features and feature conjunctions from which they are composed. This can be seen most directly in research on visual search, where participants are typically tasked with identifying a specific object within a multi-object array. A classic pattern of results arises from a comparison of two conditions: disjunctive and conjunctive search [33]. In disjunctive search (depicted on the left side of Figure 1), the array consists of distractor objects that share one feature with the target (e.g., the distractors are all circles) but differ in a second feature (e.g., the distractors are all *red* circles in the 2D task variant). Since one of the feature values (the color green) is uniquely assigned to the target object, the distractors present little interference and therefore task performance is invariant to the number of distractors. This condition is therefore sometimes referred to as "popout" search, as the target immediately stands out from the distractors, and the task can thus be performed rapidly without the need for serial processing. Conversely, in conjunctive search (depicted in the middle of Figure 1), there are two types of distractor objects that each share one feature with the target (e.g., half of the distractors are red L-shapes and the other half are green T-shapes). In this case, the target (a green L-shape) possesses no unique feature that easily distinguishes it from the distractors, leading to a significant degree of interference between the distractors and the target. One way to mitigate this is the use of serial search to identify the target. This is suggested by ubiquitously observed increases in reaction time as a function of the number of distractors, as well as the observation that when participants are prevented from engaging in serial search (e.g., by forcing participants to respond quickly), task performance degrades rapidly as more objects are added to the scene.

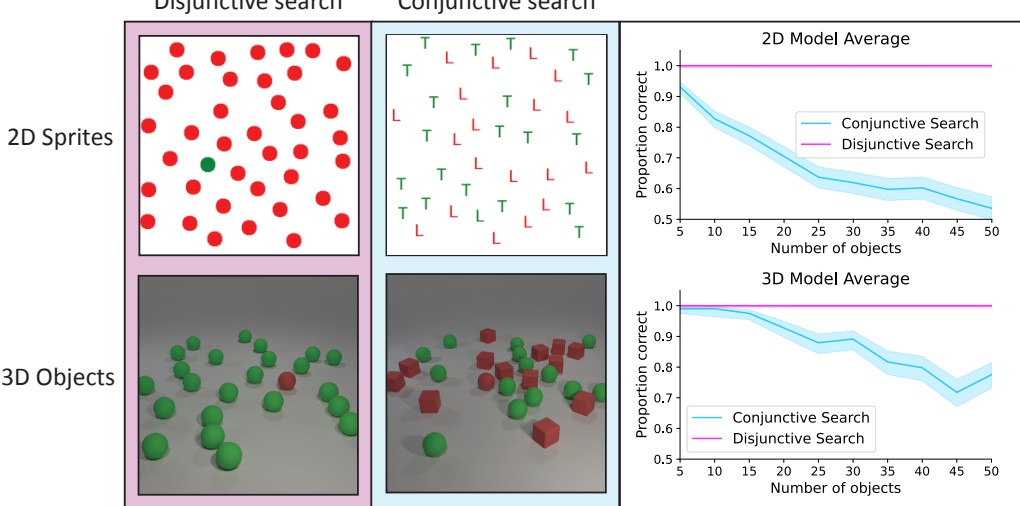

Figure 1: **Visual search tasks and results.** Example trials for the 2D (top) and 3D (bottom) variants of the disjunctive (left/red column) and conjunctive (middle/blue column) search conditions. Performance for 2D and 3D task variants are plotted on the right. Results reflect aggregate performance for all four VLMs (GPT-4v, GPT-4o, Gemini Ultra 1.5, and Claude Sonnet 3.5; see Supplementary Figure 5 for separate model results). Error bars denote 95% binomial confidence intervals.

## 2.1 Methods

We tested the extent to which VLMs demonstrate similar capacity constraints to humans in visual search tasks. We evaluated four multimodal language models – GPT-4v, GPT-4o, Gemini Ultra 1.5, and Claude Sonnet 3.5 – on a task involving disjunctive and conjunctive search conditions.[1] We generated datasets involving either 2D sprites or 3D scenes created in Blender [6] (similar to those found in the CLEVR dataset [13]). The datasets were designed to evaluate the ability of the model to detect the presence of a target object among multiple distractors. In half of the images, a target was present, while in the other half, no target was present.

Each image contained between 4 and 50 distractors. For the disjunctive search task, these consisted of non-overlapping red circles (for the 2D dataset) or green spheres (for the 3D dataset) of a uniform size. Half of the images additionally contained a target object, which was a green circle (for the 2D dataset) or a red sphere (for the 3D dataset). For the conjunctive search task, the 2D dataset consisted of images in which the distractors were either red L-shapes or green T-shapes (randomly selected with equal probability). Half of the images additionally contained a target object, which was a green L-shape. The 3D dataset consisted of images in which the distractors were either green spheres or red cubes (randomly selected with equal probability). Half of the images additionally contained a target object, which was a red sphere. Each of the datasets (2D disjunctive, 2D conjunctive, 3D disjunctive, 3D conjunctive) contained 1000 images.

## 2.2 Results

We measured the performance of each model by calculating, for each condition, how detection accuracy varied as a function of the number of distractors[2]. The results indicate that performance in the disjunctive search (i.e., popout) condition was perfect, and invariant to the number of distractors. That is, regardless of the number of distractors, all models achieved perfect accuracy in this condition.

---

[1]We also evaluated an open-source multimodal language model – Llava 1.5 – but performance was very low for these tasks. These results are presented in Supplementary Figure 6.

[2]When studying visual search in human participants, a reaction time (RT) paradigm is typically employed, measuring the time required to locate the target object. Because RT measures cannot be straightforwardly obtained from VLMs, we instead measure accuracy, which is sometimes used in human behavioral studies that employ speeded responses. [18]

In contrast, in the conjunctive search condition performance was inversely related to the number of objects: for 5 objects, all models displayed an accuracy of ∼90%, but as the number of objects increased, performance dropped substantially. These results were consistently observed for both the 2D (top panel of Figure 1) and 3D (bottom panel of Figure 1) datasets. These results were also replicated in an alternative version of the disjunctive search task, in which target and distractor colors were varied between trials (Supplementary Figure 7).

The results of these experiments suggest that multimodal language models demonstrate human-like capacity constraints in their ability to perform visual search in multi-object settings. It is important to emphasize that these capacity constraints are not driven solely by the number of objects present within a scene. Like humans, these models demonstrate capacity constraints only in the task conditions that are impacted by interference between the target and distractor objects, consistent with the hypothesis that these capacity constraints arise as a consequence of the binding problem.

## 3   Numerical Estimation

To assess the generality of the human-like capacity constraints observed for VLMs in visual processing, we investigated a simple numerical estimation task (i.e., counting) that has been widely studied in cognitive psychology. Although human observers can precisely count a very large number of items when allowed to explicitly process those items one at a time, their ability to rapidly estimate the number of items in a display is subject to a severe capacity constraint. Studies have found that the number of objects that can be reliably estimated without explicit serial counting (sometimes referred to as "subitizing") is somewhere between 4 and 6 [14, 17, 19, 27, 34]. To determine whether VLMs are subject to similar constraints, we evaluated both multimodal language models (GPT-4v, GPT-4o, Gemini Ultra 1.5, Claude Sonnet 3.5, and Llava 1.5) and text-to-image models (Stable Diffusion Ultra, DALL-E 3, Google Parti, and Google Muse) on a numerical estimation task involving variations of both the number and type of objects. We found that VLMs, across a variety of stimulus and model types, display strikingly similar quantitative capacity limits to those observed in human vision. We also found that these capacity constraints were strongly affected by the variability of features present in an image. This effect is consistent with the hypothesis that these constraints arise due to representational interference: given that objects are represented with a shared set of representational resources, greater feature variability leads to less overlap in the use of these resources, and therefore less opportunities for interference and binding errors.

### 3.1   Methods

We generated datasets involving both 2D sprites and 3D objects, varying the number of objects per image between 1 and 20. We explored four conditions with varying levels of feature entropy (i.e., feature variability): a low-entropy condition in which all objects in an image had the same color and shape; two medium-entropy conditions in which all objects in an image had the same shape but unique colors, or vice versa; and a high-entropy condition in which all objects in an image had unique colors and shapes. We prompted the multimodal language models to describe the image and then state the number of objects present in it. To test the text-to-image models, we generated a dataset comprising 100 distinct categories, evenly split between common foods (50 categories) and animals (50 categories). We tasked these models with generating images from each category, for which the number of instances of each object ranged from 1 to 10. To assess their ability to generate images with the exact number of objects requested, we conducted a human evaluation study. Participants were asked to count and report the number of objects visible in each generated image. The collected human judgments were then used to quantify the model's accuracy. See Appendix C for further details on human evaluations.

### 3.2   Results

We measured performance by calculating, for each condition, how accuracy varied with the number of objects present in the scene. The results indicated that, regardless of the type of stimuli used (2D vs. 3D shapes, or animals vs. food), and across two fundamentally different types of vision language model (multimodal language models and text-to-image models), VLMs displayed human-like capacity limits (Figure 2). For both multimodal language models and T2I models, accuracy was very high for scenes involving a relatively small number of objects (1-5), but dropped sharply for scenes involving

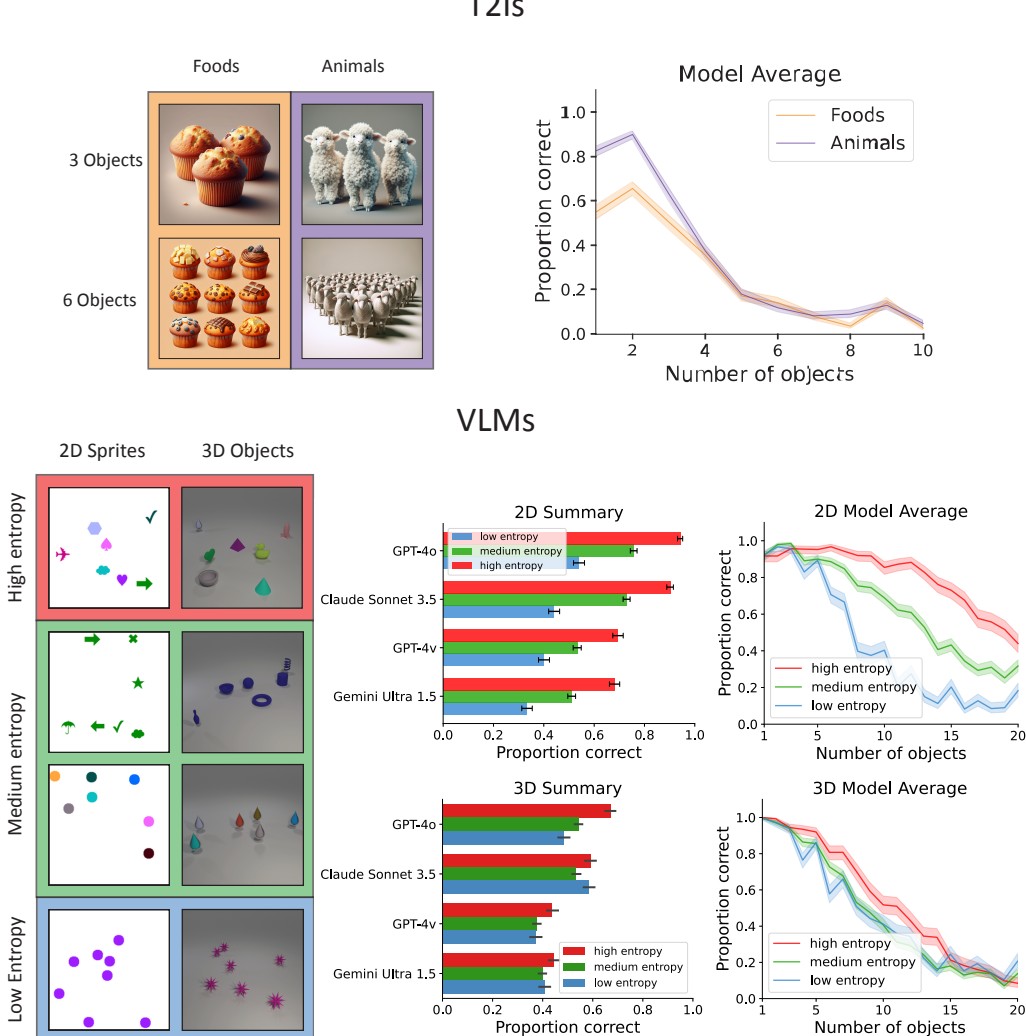

Figure 2: **Numerical estimation tasks and results.** Top left: Examples of images generated by text-to-image (T2I) models for different numbers and categories of objects. Top right: Performance of T2I models as a function of the number and category of objects. Results reflect an aggregate of four models (Stable Diffusion Ultra, DALL-E 3, Google Parti, and Google Muse). Bottom left: Examples of images (featuring either 2D or 3D objects) used to evaluate numerosity estimation. Feature entropy was varied in four conditions (low entropy, high entropy, and two medium entropy conditions). Bottom middle: Numerosity estimation results for four multimodal language models (GPT-4v, GPT-4o, Gemini Ultra 1.5, Claude Sonnet 3.5; see Supplementary Figure 6 for results with Llava-1.5). Bottom right: Numerosity estimation results plotted as a function of the number of objects in an image, aggregated across all four models (see Supplementary Figure 8 for individual model results). Error bars for all plots reflect 95% binomial confidence intervals.

6 or more objects. Moreover, the multimodal language models exhibited performance consistent with our hypothesis that capacity limits arise due to representational interference across objects (i.e., the binding problem), with overall performance highest in the high-entropy condition (lowest interference), lowest in the low-entropy condition (highest interference), and intermediate in between these two extremes in the medium-entropy conditions. Though there are slight differences between the capacity limits exhibited by these two classes of models, it is striking that they both fall within the subitizing limit of human vision, especially when considering the significant differences in both architecture and training procedures. Furthermore, the effect of feature entropy on these capacity

limits strongly suggests that they are driven by representational interference, arising due an inability to manage the binding problem. To further investigate this hypothesis, we next turned to a novel scene description benchmark that allowed us to systematically investigate the likelihood of representational interference, thus enabling a direct test of the extent to which this factor is responsible for the shortcomings of VLMs.

## 4 Scene Description

Theoretical accounts of the binding problem [9, 33] posit that capacity limits in rapid visual processing arise as a consequence of interference between representations. Given a scene containing multiple objects, and a set of shared features with which to represent those objects, the likelihood of interference will tend to increase as a function of the number of objects in the scene (without the availability of a mechanism for binding features together, e.g., serial processing). However, as emphasized in our experiments on visual search and numerosity estimation, interference is not driven solely by the number of objects, but is also strongly influenced by the specific feature conjunctions present within a scene.

We developed a novel scene description task to further investigate the extent to which VLM performance is driven by representational interference. The task is illustrated in Figure 3a. For each image, the likelihood of representational interference was quantified as the number of *feature triplets* present in that image. A feature triplet is defined as any set of three objects for which one pair shares a feature, and another pair shares a different feature. For instance, {green X, green triangle, yellow triangle} is a feature triplet, because the feature 'green' is shared by two objects (the green X and the green triangle), and the feature 'triangle' is shared by two objects (the green triangle and the yellow triangle). Without the ability to accurately bind these features together at the level of objects, such feature triplets create opportunities for representational interference, and thus lead to illusory conjunctions. For instance, the feature triplet {green X, green triangle, yellow triangle} may lead to the erroneous identification of a yellow X. We studied the extent to which the presence of such feature triplets can account for scene description performance in VLMs.

### 4.1 Methods

As in the previous tasks, we generated datasets involving either 2D sprites or 3D objects. Each scene contained a variable number of objects (10-15 objects for the 2D dataset and 8-12 objects for the 3D dataset), and we systematically varied the number of feature triplets present in each scene. For example, the scene depicted in Figure 3a contains three feature triplets (illustrated by the dashed lines). VLMs (GPT-4v, GPT-4o, Gemini Ultra 1.5, and Claude Sonnet 3.5) were prompted to provide a description of the objects in JSON format (see Appendix B for more details). We also generated prompts describing similar scenes (but involving real-world objects) and tested the ability of the T2I models (Stable Diffusion Ultra, DALL-E 3, Google Parti, and Google Muse) to accurately generate these scenes (as assessed by human evaluation; see Appendix C for more details). To obtain a representative sampling of scenes with different triplet counts, we systematically varied the diversity of colors and shapes across trials. This approach ensured adequate sampling of trials with different feature combinations and their associated triplet counts. To ensure reliable performance estimates, we excluded from analysis any triplet counts represented by fewer than 20 trials across all conditions. For the T2I experiments, we additionally excluded trials where models generated more than three extraneous objects not specified in the prompt, as these represented significant deviations from the intended scene structure.

### 4.2 Results

We measured scene description performance by calculating how the number of errors (quantified as the edit distance between the true description of the scene and the model's description of the scene) varies as a function of the number of objects present in the scene, and the number of feature triplets. The results (Figure 3) confirmed our prediction that performance should vary as a function of the number of triplets. Across multiple stimulus types (2D and 3D objects), and model types (both multimodal language models and text-to-image models), the largest number of errors occurred in the trials where the risk of binding errors was highest (i.e., the trials with the largest number of feature triplets), consistent with the hypothesis that errors would be driven primarily by the formation

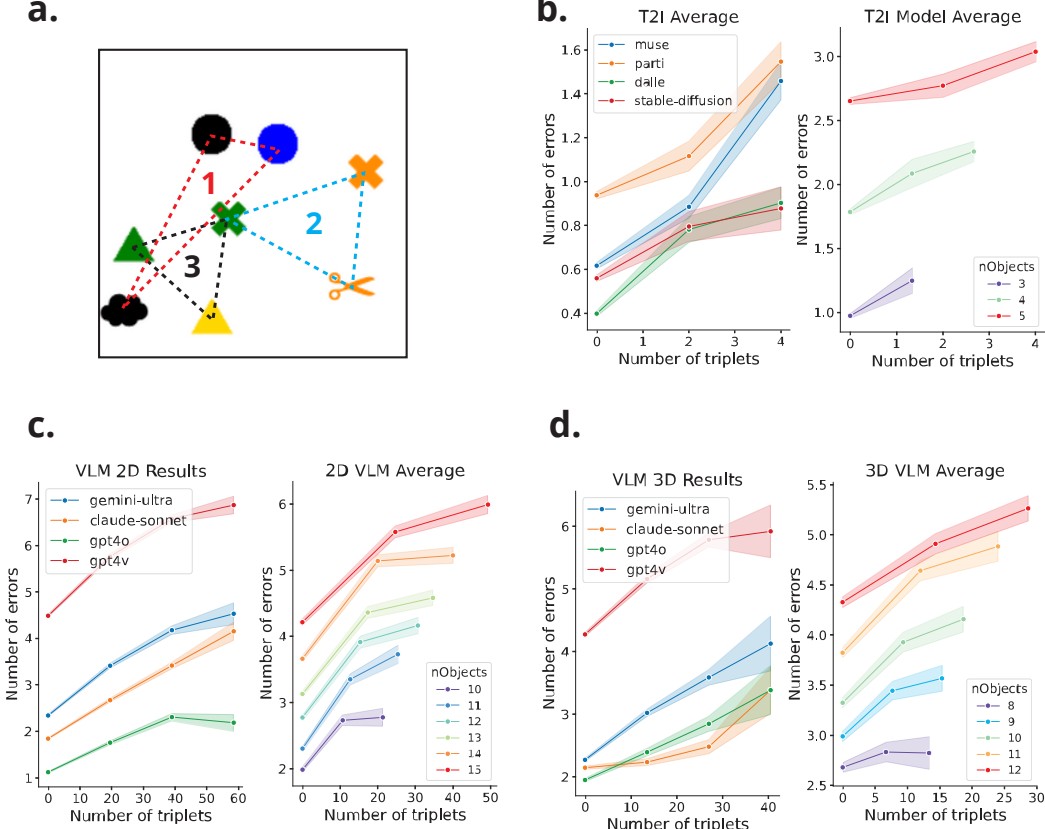

Figure 3: **Scene description task and results.** A) Example image used in 2D scene description task, illustrating the concept of *feature triplets*: sets of three objects where one pair of objects shares a feature, and another pair shares a different feature. This example contains three feature triplets, demarcated by the dashed lines. 3D scenes were also investigated. B) Scene description results for text-to-image (T2I) models) as a function of the number of feature triplets. C) 2D scene description results for multimodal language models as a function of the number of feature triplets. Left panel illustrates the results aggregated across four models (GPT-4v, GPT-4o, Gemini Ultra 1.5, and Claude Sonnet 3.5). Right panel illustrates the results aggregated across scenes with different numbers of objects. D) 3D scene description results. Error bars represent the standard error of the mean.

of illusory conjunctions. These results also showed a pattern of increasing errors as a function of the number of objects, consistent with a general capacity limit on the number of objects that can be accurately represented at the same time. Overall, these results suggest that the capacity limits displayed by VLMs are best explained by an inability to manage the binding problem. In the next section, we apply these insights to better understand the limited visual reasoning capabilities of VLMs.

## 5 Visual Analogy

An open question in studying the performance of VLMs is the extent to which these models can solve analogical reasoning tasks. These tasks are of particular interest given their centrality in human higher-order cognition [11] and their use as measures of human intelligence [30]. Recent work has demonstrated that LLMs have an impressive ability to solve a range of text-based analogical reasoning tasks [37], but initial tests of VLMs have suggested that they often struggle to solve comparable visual forms of these tasks, sometimes performing well below human participants [20, 38].

This leads to the question of why, given the success of LLMs on text-based problems in this domain, VLMs do not display comparable success in solving analogy tasks. One possible explanation,

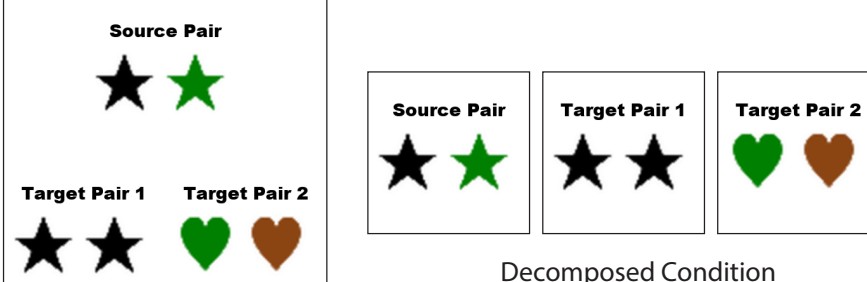

Figure 4: **Visual analogy task.** The Unified and Decomposed conditions present the same object pairs, but in the Decomposed condition it is broken up across three images. The correct target pair must share both relations (shape and color) with the source pair, so the correct answer in this example is Target Pair 2 because it satisfies both the 'same shape' and 'different color' relations.

suggested by our investigation of the binding problem, is that the difficulty displayed by VLMs on visual reasoning tasks may stem from a more basic difficulty with processing multi-object scenes. In other words, regardless of whether VLMs have the capacity for abstract reasoning necessary to solve analogy tasks, they will likely struggle to solve *visual* analogy tasks simply because these tasks involve processing multi-object scenes. To distinguish between these two failure modes (inability to process abstract relations vs. inability to process multi-object scenes), we performed experiments using a visual analogy task in which the visual processing demands were explicitly manipulated, and measured the ability of multimodal language models to perform both abstract relational tasks and basic object-level tasks.

## 5.1   Methods

We generated 200 trials from a simple relational match-to-sample (RMTS) task (Figure 4) using the same 2D sprite stimuli from the previous experiments. We selected a subset of 8 easily recognizable shapes and colors to generate a set of 64 stimuli. For each trial, we chose the source pair by sampling two objects that shared at least one of the two feature dimensions. We then selected the target pairs by sampling two pairs of objects: one which matched the source pair exactly along its relations (the correct target) and one which shared only one of the relations with the source pair (the incorrect target). We manipulated visual processing demands by investigating two conditions, one in which the source and target pairs were presented in a single image (the "unified" condition), and one in which the source and target pairs were presented as separate images presented in sequence (the "decomposed" condition), thereby reducing the chance of binding errors.

We assessed the performance of four VLMs (GPT-4v, GPT-4o, Gemini Ultra 1.5, and Claude Sonnet 3.5) in four tasks: identification of the correct target pair in the full RMTS task (Analogy), decoding of single features from specific individual objects (Single Feature Decoding Task), comprehensive decoding of all features in a given problem (Full Feature Decoding), and decoding of relations between object pairs (Relation Decoding).

## 5.2   Results

We found that performance on this task was highly variable aross VLMs, with some models (Claude Sonnet) showing nearly perfect performance on all tasks (Table 3) and other models (Gemini Ultra) showing poor performance on most tasks (Table 4). These results are consistent with recent work showing mixed success on visual analogy problems [38], and at odds with work claiming that VLMs have no capacity for visual analogy [20]. Interestingly, we also found that many models also struggled on more basic tasks such as identifying the features of the objects present in the image, or identifying the relations for individual pairs of objects.

Table 1: **Visual analogy results: GPT-4v.**

|  | Unified Accuracy | | Decomposed Accuracy | |
|---|---|---|---|---|
|  | Accuracy | 95% CI | Accuracy | 95% CI |
| Analogy | 91% | (86%, 94%) | **99%** | (97%, 99%) |
| Relation decoding | 80% | (73%, 84%) | **91%** | (86%, 94%) |
| Full feature decoding | 85% | (79%, 89%) | **100%** | (100%, 100%) |
| Single feature decoding | 95% | (92%, 98%) | 98% | (96%, 99%) |

Table 2: **Visual analogy results: GPT-4o.**

|  | Unified Accuracy | | Decomposed Accuracy | |
|---|---|---|---|---|
|  | Accuracy | 95% CI | Accuracy | 95% CI |
| Analogy | 99% | (96%, 100%) | 100% | (100%, 100%) |
| Relation decoding | 88% | (82%, 92%) | **98%** | (95%, 99%) |
| Full feature decoding | 97% | (84%, 100%) | 100% | (100%, 100%) |
| Single feature decoding | 86% | (81%, 91%) | **94%** | (81%, 91%) |

Table 3: **Visual analogy results: Claude Sonnet 3.5.**

|  | Unified Accuracy | | Decomposed Accuracy | |
|---|---|---|---|---|
|  | Accuracy | 95% CI | Accuracy | 95% CI |
| Analogy | 100% | (100%, 100%) | 100% | (100%, 100%) |
| Relation decoding | 92% | (88%, 95%) | **99%** | (97%, 100%) |
| Full feature decoding | 100% | (100%, 100%) | 100% | (100%, 100%) |
| Single feature decoding | 100% | (100%, 100%) | 100% | (100%, 100%) |

Table 4: **Visual analogy results: Gemini Ultra 1.5.**

|  | Unified Accuracy | | Decomposed Accuracy | |
|---|---|---|---|---|
|  | Accuracy | 95% CI | Accuracy | 95% CI |
| Analogy | 56% | (49%, 64%) | 60% | (53%, 67%) |
| Relation decoding | 84% | (79%, 89%) | 89% | (83%, 93%) |
| Full feature decoding | 100% | (100%, 100%) | 100% | (100%, 100%) |
| Single feature decoding | 81% | (74%, 85%) | 83% | (78%, 88%) |

Most importantly, we found that performance was significantly improved in the Decomposed condition (involving separate images for each pair of objects) as compared with the Unified condition (involving a single image with all object pairs). This was the case across all tasks, and for all models, except for the cases in which performance was already at ceiling for the Unified condition. Taken together, these results suggest that the poor performance of VLMs on visual analogy tasks may be a consequence of a more general limitation with processing multi-object scenes, arising due to an inability to manage the binding problem.

## 6   Discussion

We have presented a series of experiments aimed at understanding the limits of vision language models in processing multi-object scenes. Our results suggest that these limitations can all be understood as arising from an inability to manage the binding problem, a fundamental problem associated with compositional coding identified by classic work in cognitive science [10, 33].

Recent theoretical work has formalized this problem within a normative framework [9], suggesting that it arises due to a tension between the learning of compositional representations, and the shared use of such representations to encode multiple objects at the same time. To illustrate this, consider two different schemes for representing multi-object scenes: a *conjunc-*

*tive* scheme, involving dedicated representations for every possible conjunction of features (e.g., {redsquare, greencircle, bluetriangle, greensquare, . . .}) vs. a *compositional* scheme, involving the dynamic combination of shared features across different dimensions (e.g., combining the features {red} and {triangle} to represent a red triangle). The compositional scheme enables efficient use of finite neural resources, and offers major benefits in terms of generalization (e.g., anything learned about red triangles can then be readily generalized to support inferences about other red objects), but, without a mechanism for dynamically keeping track of the bindings between features, this scheme leads to severe interference, giving rise to the capacity limits observed in cognitive processing. One surprising implication of this perspective is that the binding errors exhibited by VLMs suggest that they have developed compositional representations, perhaps as a consequence of being forced to generalize by their immense and highly diverse training corpora. Without the use of such compositional representations (i.e., if VLMs employed a conjunctive coding scheme), there would be no interference between the representations for different objects, and thus there would be no binding problem. The presence of the binding problem, therefore, implies the presence of compositional representations in VLMs.

It is worth considering how VLMs might be improved so as to enable them to cope with the binding problem. One might think that this could be accomplished through additional fine-tuning on multi-object tasks. However, the theoretical perspective outlined above suggests that, to the extent this alleviates the binding problem, it would do so by eliminating the use of compositional representations, which would then have negative impacts on generalization. The question then is how VLMs might be enhanced to solve the binding problem, while *preserving the benefits of compositional representations*. The most obvious possibility here is to augment VLMs with mechanisms for serial processing of images, of the sort that enable human reasoners to manage the binding problem by selectively attending to individual objects one at a time [22, 33]. A number of methods have been proposed for sequential reasoning over images [12, 35], though none of these methods have yet been deployed at the scale of VLMs. An alternative approach (which may be unique to artificial systems) involves the use of slot-based methods for object-centric representation learning [4, 10, 16], which have been shown to dramatically improve performance in visual reasoning tasks without requiring sequential processing of images [8, 21], but which have also not been scaled to the level of current VLMs. It remains to be seen whether and how these techniques might contribute to improved reasoning in future VLMs, or whether new approaches will be needed to enable human-like visual reasoning.

## 6.1 Limitations & Future Directions

This study has several limitations that should be considered when interpreting the results. First, we limited our analysis to a relatively small set of tasks. The tasks in our study were selected to illustrate the different settings in which the binding problem may impact performance, while grounding our analysis in well known tasks from cognitive science that have been used to index such capacity constraints in humans. Future work may examine a broader set of tasks such as matrix reasoning tasks [3, 26, 39] that are more diagnostic of the reasoning failures arising due to issues with binding. Second, we primarily investigated propietary VLMs, for which we do not have detailed knowledge of their training data or architecture, or the ability to directly investigate their internal representations. We chose to focus on these models because they reflect the best-performing current set of VLMs (our experiments with the open-source Llava-1.5 yielded very poor performance on all tasks), but continued progress in the development of open-source VLMs should make it possible to investigate open-source models in future work. Finally, our work is focused particularly on characterizing the capacity constraints of VLMs arising due to issues with feature binding. While we propose a naive approach for improving performance by selectively processing sub-images independently, future work may explore more flexible methods for decomposing complex, multi-object reasoning tasks, especially by exploiting methods for object-centric representation learning [4, 5, 10, 16].

## 7 Acknowledgements

This work was supported in part by Microsoft Azure credits provided to Princeton University. D.C. and T.G. are supported by the National Science Foundation Graduate Research Fellowship Program (NSF GRFP).

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

# A   Appendix: Supplementary Figures

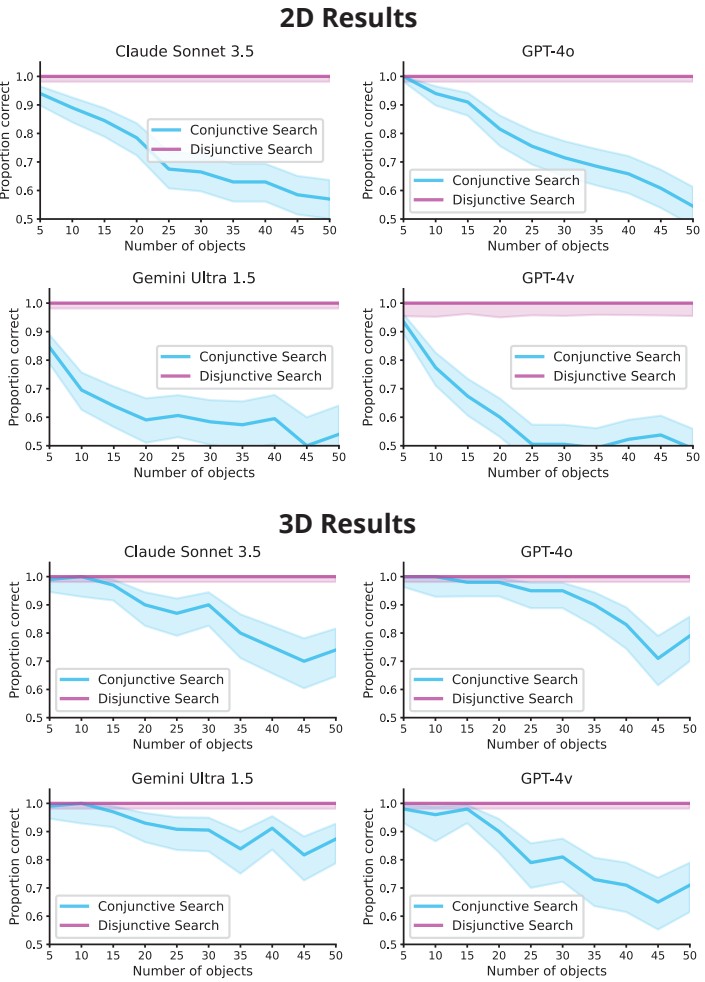

Figure 5: **Visual search model results.** Individual model performance for 2D and 3D visual search tasks. Error bars denote 95% binomial confidence intervals.

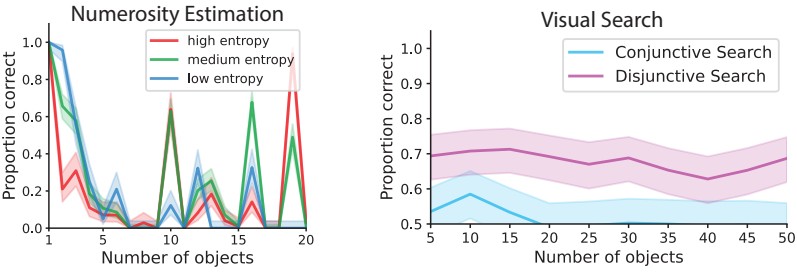

Figure 6: **LLaVA 1.5 Performance.** Performance of LLaVA 1.5 on the visual search and numerosity estimation tasks. Performance was substantially weaker than all other models evaluated.

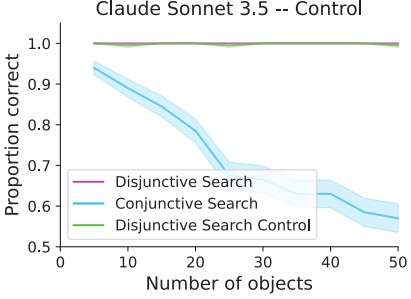

Figure 7: **Visual search results with additional control experiment.** Results for Claude Sonnet 3.5 on 2D visual search tasks, including disjunctive and conjunctive conditions, and an additional disjunctive search condition ('Disjunctive Search Control') in which target and distractor colors were varied between trials.

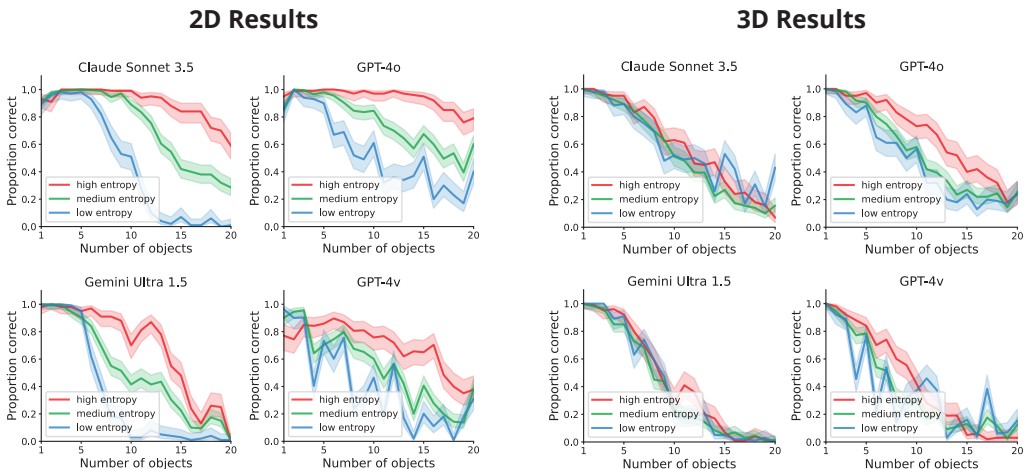

Figure 8: **Numerosity estimation model results**. Individual model performance for 2D and 3D numerosity estimation tasks. Error bars denote 95% binomial confidence intervals.

# B Appendix: Prompts for Vision-Language Model Experiments

## B.1 Numerical Estimation

1. **2D VLM Prompt**

```
You are presented with an image containing several objects. Your task is to
   accurately count the number of objects in the image. Follow these
   instructions carefully:
1. Begin by describing each object in the image.
2. Conclude your response by providing the total count of objects as an integer
   , enclosed in square brackets. Only the number should be enclosed in square
   brackets.
```

2. **3D VLM Prompt**

```
You are presented with an image containing several shapes. Your task is to
   accurately count the number of shapes in the image. Follow these instructions
   carefully:
1. Begin by describing each shape in the image.
2. Conclude your response by providing the total count of shapes as an integer,
   enclosed in square brackets. Only the number should be enclosed in square
   brackets.
```

3. **T2I Prompt**

```
Render an image with exactly {n} {object_name}, each distinctly separated and
   easily countable, arranged against a uniform background in photorealistic
   style.
```

## B.2 Visual Search

1. **2D Disjunctive Search Prompt**

```
You are presented with an image containing several shapes. Your task is to
   determine if there are any green shapes in the image. Follow these steps
   carefully:
1. Describe each shape in the image, noting their color.
2. Conclude your response by stating [True] if there are any green shapes, or [
   False] if there are none. Enclose your final answer in square brackets, as
   shown.
```

2. **3D Disjunctive Search Prompt**

```
You are presented with an image containing several objects. Your task is to
   determine if there are any red objects in the image. Follow these steps
   carefully:
1. Describe each object in the image, noting their color.
2. Conclude your response by stating [True] if there are any red objects, or [
   False] if there are none. Enclose your final answer in square brackets, as
   shown.
```

3. **2D Disjunctive Search (Variable Color) Prompt**

```
You are presented with an image containing several shapes. Your task is to
   determine if all the shapes are the same color. Follow these steps carefully:
1. Describe each shape in the image, noting their color.
2. Conclude your response by stating [True] if all shapes are the same color,
   or [False] if there is an "oddball" shape that is a different color. Enclose
   your final answer in square brackets, as shown.
```

4. **2D Conjunctive Search**

```
You are presented with an image containing a set of letters, specifically the
   letters 'L' and 'T'. These letters will appear in either red or green.
Your task is to determine if there are any green 'L's in the image. Follow
   these steps carefully:
1. Describe each shape in the image, noting their color.
2. Conclude your response by stating [True] if the letter 'L' appears in green,
   or [False] if there are no green 'L's. Enclose your final answer in square
   brackets, as shown.
```

5. **3D Conjunctive Search Prompt**

```
You are presented with an image containing a set of objects, specifically
    spheres and cubes. These objects will appear in either red or green.
Your task is to determine if there are any red spheres in the image. Follow
    these steps carefully:
1. Describe each object in the image, noting their color.
2. Conclude your response by stating [True] if a red sphere is present, or [
    False] if there are none. Enclose your final answer in square brackets, as
    shown.
```

## B.3 Scene Description

### 1. 2D VLM Prompt

```
The following image contains multiple simple, colored objects.
The possible shapes that may be present in the image are: <airplane, triangle,
    cloud, X-shape, umbrella, pentagon, heart, star, circle, square, spade,
    scissors, infinity, check mark, right-arrow>.
The possible colors that may be present in the image are: <red, magenta, salmon
    , green, lime, olive, blue, teal, yellow, purple, brown, gray, black, cyan,
    orange>.
Describe each object in the image in the form of a JSON object detailing the
    color and shape of each item.
You must answer only with the json array of objects, without any additional
    information or text.
For example, if the image contains a purple check mark, two green scissors, one
     orange right-arrow, and a teal infinity sign you would write:

[
    {"shape": "check mark", "color": "purple"},
    {"shape": "scissors", "color": "green"},
    {"shape": "scissors", "color": "green"},
    {"shape": "right-arrow", "color": "orange"},
    {"shape": "infinity", "color": "teal"}
]
```

### 2. 3D VLM Prompt

```
The following image contains multiple simple, colored objects.
The possible shapes that may be present in the image are: <cone, cylinder, bowl
    , donut, sphere, cube, droplet, bowling-pin, coil, crown, snowman, spikey-
    ball>.
The set of colors that may be present in the image are: <red, green, blue,
    yellow, purple, light green, gray, black, light blue, pink, teal, brown>.
Describe each object in the image in the form of a JSON object, detailing the
    color and shape of each item.
You must answer only with the json array of objects, without any additional
    information or text.
For example, if the image contains a brown cube, two green donuts, and a cyan
    spikey-ball, you would write:

[
    {"shape": "cube", "color": "brown"},
    {"shape": "donut", "color": "green"},
    {"shape": "donut", "color": "green"},
    {"shape": "spikey-ball", "color": "cyan"}
]
```

### 3. T2I Prompt

```
Render an image in photorealistic style with exactly {objects_string} arranged
    against a uniform background, each distinctly separated. Include only these
    objects in the image and nothing else.
```

## B.4 Relational Match to Sample (RMTS)

### 1. Full Task – Unified Condition

```
The following image depicts a trial of a relational match to sample task with
    two features: shape and color.
There are three pairs of objects relevant to your task: the source pair (the
    top pair of objects), target pair #1 (the pair of objects on bottom left),
    and target pair  (the pair of objects on the bottom right).
Now, given the source pair and the two target pairs, identify the matching
    target pair. To accomplish this task, you can use the following steps:
1. Identify the features of the objects in each pair (i.e. shape and color).
2. Identify the relations over the features of each pair.
3. Determine which target pair shares the same relations with the source pair
    -- exactly one target pair will share the same relations with the source.
4. Conclude with the integer of the correct target pair wrapped in square
    brackets. For example, if target pair #1 matches the source pair, return [1].
```

### 2. Full Task – Decomposed Condition

```
The following images depict a trial of a relational match to sample task with
    two features: shape and color.
There are three pairs of objects relevant to your task: the source pair, target
     pair #1, and target pair #2.
Now, given the source pair and the two target pairs, identify the matching
    target pair. To accomplish this task, you can use the following steps:
1. Identify the features of the objects in each pair (i.e. shape and color).
2. Identify the relations over the features of each pair.
3. Determine which target pair shares the same relations with the source pair
    -- exactly one target pair will share the same relations with the source.
4. Conclude with the integer of the correct target pair wrapped in square
    brackets. For example, if target pair #1 matches the source pair, return [1].
```

### 3. Relation Decoding – Unified condition

```
Are the two objects in the {pair} pair (the {pair_loc} pair) the same {relation
    }?
Your answer should be [True] if the objects have the same {relation} and [False
    ] if they have a different {relation}.
Ensure that your final answer is wrapped in square brackets.
```

### 4. Relation Decoding – Decomposed condition

```
Are the two objects in the {pair} pair (the {pair_loc} pair) the same {relation
    }?
Your answer should be [True] if the objects have the same {relation} and [False
    ] if they have a different {relation}.
Ensure that your final answer is wrapped in square brackets.
```

### 5. Single Feature Decoding – Unified condition

```
What is the {feature} of the {object_loc} ({object_ind}) object in the {pair}
    pair? Only provide the {feature}.
Your response should be a single word answer wrapped in square brackets.
For instance, if I ask for the color of a red object, you should return [red]
    and nothing else. If I ask for the shape of an object that is a circle, you
    should return [circle].
- Valid shapes include: triangle, cloud, cross, heart, circle, square.
- Valid colors include: red, green, blue, darkorange, purple, and gray.
```

### 6. Single Feature Decoding – Decomposed condition

```
What is the {feature} of the {object_loc} ({object_ind}) object in the {pair}
    pair? Only provide the {feature}.
Your response should be a single word answer wrapped in square brackets.
For instance, if I ask for the color of a red object, you should return [red]
    and nothing else. If I ask for the shape of an object that is a circle, you
    should return [circle].
- Valid shapes include: triangle, cloud, cross, heart, circle, square.
- Valid colors include: red, green, blue, darkorange, purple, and gray.
```

## 7. All Feature Decoding – Unified condition

```
Examine the image provided, which depicts six basic, colored shapes arranged
  into three distinct pairs of objects: the source pair at the top, target pair
  #1 on the bottom left, and target pair #2 on the bottom right.
For each pair, identify the shapes as follows: "object1" refers to the left-
  most object in the pair, and the "object2" to the right-most object in the
  pair.
Return the color and shape of each object in the trial in the json format
  described below.
- Valid shapes: triangle, cloud, cross, heart, circle, square.
- Valid colors: red, green, blue, darkorange, purple, and gray.

Your response should be in the following format:
{
    source: {
      source_object1: {shape: circle, color: purple},
      source_object2: {shape: circle, color: purple}
    },
    target1: {
      target1_object1: {shape: triangle, color: brown},
      target1_object2: {shape: triangle, color: brown}
    },
    {
      target2_object1: {shape: square, color: green},
      target2_object2: {shape: square, color: black}
    }
}

Response:
```

## 8. All Feature Decoding – Decomposed condition

```
Examine the three images provided, which depict six basic, colored shapes
  arranged into three distinct pairs of objects: the source pair, target pair
  #1, and target pair #2.
For each pair, identify the shapes as follows: "object1" refers to the left-
  most object in the pair, and the "object2" to the right-most object in the
  pair.
Return the color and shape of each object in the trial in the json format
  described below.
- Valid shapes: triangle, cloud, cross, heart, circle, square.
- Valid colors: red, green, blue, darkorange, purple, and gray.

Your response should be in the following format:
{
    source: {
      source_object1: {shape: circle, color: purple},
      source_object2: {shape: circle, color: purple}
    },
    target1: {
      target1_object1: {shape: triangle, color: brown},
      target1_object2: {shape: triangle, color: brown}
    },
    {
      target2_object1: {shape: square, color: green},
      target2_object2: {shape: square, color: black}
    }
}

Response:
```

# C   Appendix: Human Evaluations

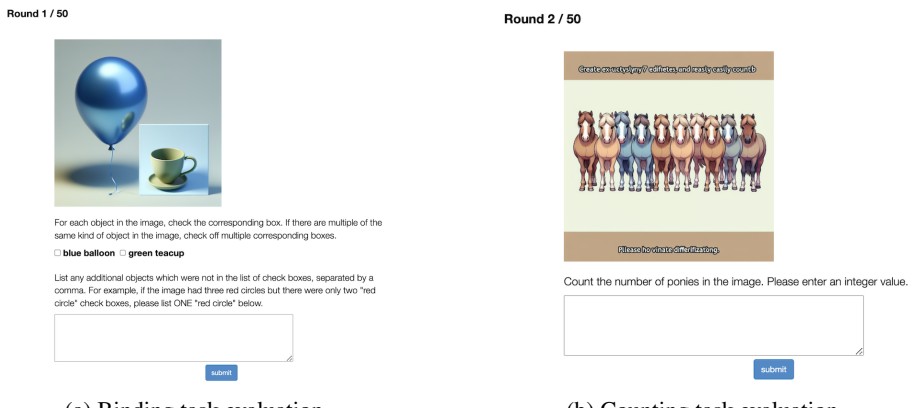

(a) Binding task evaluation    (b) Counting task evaluation

Figure 9: **Human Evaluation**

Human evaluations of the text-to-image variants of the counting and binding tasks were conducted on Prolific. Participants were asked to count the number of objects in each T2I counting task image, or asked to match objects to provided labels in each T2I scene description task image. They were also asked to list extraneous objects (generated objects not in the input prompt) when evaluating the binding images. Participants were paid a total estimated wage of approximately $12/hour, and total compensation for the entirety of the human evaluations was approximately $600. All participants provided informed consent. The study was approved by the Princeton University IRB.

# D   Appendix: Computational Resources

Experiments were conducted on closed source models, and therefore did not require any specialized hardware. Generation of the 3D variants of the task were more computationally intensive and were performed in parallel on a local cluster, with 16GB RAM allocated per CPU per job.

