# OpenReview forum: "Understanding the Limits of Vision Language Models Through the Lens of the Binding Problem"
_NeurIPS.cc/2024/Conference — NeurIPS 2024 poster_

### Official Review · Reviewer_Ga5j · 2024-07-04

**Soundness:** 4
**Presentation:** 3
**Contribution:** 3
**Rating:** 8
**Confidence:** 4

**Summary:**

Vision Language Models (VLMs), including models that are able to describe images, such as GPT-4v, and models that generate images given a text description, such as DALL-E, display several failures in tasks that humans are able to perform effortlessly. Such tasks include counting, localizing a given object in an image, and visual analogy. This paper interprets these failures as resulting from a trade-off between representational flexibility (the ability to generalize compositionally to novel combinations of features) and channel capacity (the number of entities that can be represented simultaneously). This trade-off corresponds to a well-established phenomenon in cognitive science, the binding problem. The binding problem identifies a series of failures in human visual perception, that arise in many situations in which human subjects need to process multi-object scenes under time constraints. Such constraints prevent them from processing those scenes in a sequential way, and force them to rely on parallel processing, which incurs into capacity limits. This leads to phenomena such as illusory conjunctions, in which a feature is bound to the wrong object: for example, a green triangle might be falsely detected in a scene comprising red triangles, green circles and red circles. The failures shown by VLMs are remarkably similar, suggesting that they might result from the lack of a serial processing mechanism like the one humans can rely on in the absence of time constraints. To test this hypothesis, in this work VLMs are tested on two classic cognitive tasks (counting and visual search), showing error patterns similar to time-constrained human subjects. In counting, a sharp drop in performance is observed with scenes comprising more than 4 to 6 objects, a similar range to that beyond which humans are unable to “subitize” (reliably count objects in parallel, without serial processing). In visual search, conjunctive search, in which a unique combination of features needs to be detected, led to a sharp drop in performance with increasing numbers of objects, while disjunctive search, in which targets can be detected based on a single feature, did not. These results are remarkably similar to the symptoms of the binding problem in humans. The authors further posit that these failures result from interference between target and distractor objects, and not by the number of objects alone. They test this hypothesis by parametrically generating scenes that vary in their feature entropy: the amount of variation in the two features, shape and color, across all objects. Using a scene description task, they find that errors were most frequent when the feature entropy was intermediate, leading to a combination of diversity in features and sharing of features between objects. Finally, the VLMs are tested on a visual analogy task. The known failures of these models in solving these tasks is hypothesized to also depend on limitations in the number of objects that can be processed in parallel. To alleviate this problem, the authors propose a simple “decomposed condition”: feeding the target stimuli (pairs of objects) to the models in sequence, rather than as part of a single image (”unified condition”). The decomposed condition mimicks the serial processing that humans adopt in similar tasks. In both the analogy task and extraction of individual object features, VLMs perform better in the decomposed condition than the unified condition, providing further evidence that VLMs’ failures are due to capacity limits in binding features to multiple objects in parallel.

**Strengths:**

- The paper presents a simple framework to explain failure cases of VLMs, directly inspired by a long tradition of research in cognitive science. It is a great example of how cognitive science research can inform the study of widespread deep learning models, with many potential applications in evaluating and improving such models.
- The research methods are sound: the link between the theoretical question and the experiments is clear, and models are tested across multiple tasks and conditions (e.g. 2D and 3D scenes, different object categories) to ensure that results are robust.
- Creative, but at the same time simple and elegant, ways to measure the relation between the binding problem and VLM failures (feature entropy) and to alleviate the binding problem in VLMs (decomposed condition) are proposed.
- The interpretation of the results is fair and clear.

**Weaknesses:**

The weaknesses are mostly related to the lack of detail concerning some of the methods, or the lack of explanations about certain choices. No major shortcoming was found by this reviewer.

- In some of the prompts used to run experiments on the models (e.g. for counting), the model is first prompted to describe every object in the image, and then to perform the task. In others (e.g. disjunctive visual search) the model is directly asked to answer the question of interest (e.g. whether a target object is present in the image or not). Moreover, prompts from some tasks (e.g. 2D conjunctive search) include a form of role-playing "You have perfect vision and pay great attention to detail...". Since it is well known that similar techniques can have a great impact on the quality of language models' responses, the authors should provide an explanation of why these techniques were used in certain conditions and not others. Otherwise, performance comparisons between conditions (e.g. conjunctive vs. disjunctive search) can become harder to interpret.
- In the scene description task, a language model is used to parse the descriptions. These are, I suppose, harder to parse than the simple answers provided in other tasks, such as the number of objects in the counting task or True or False in the visual search task. The authors should show that the language model was accurate in parsing these descriptions. Manually validating all of them would be infeasible, but at least a few examples to show the language models' accuracy should be provided in the supplementary materials.
- Also in the scene description task, the generation of scenes with different feature entropies should be explained in more detail. Did the feature entropy manipulation simply consist in setting a particular number of shapes and features, and assigning them to a fixed number of objects? Or was the feature entropy computed in a different way? Making this clear would also help interpreting the values on the feature entropy that are shown in Figure 3.
- In Figure 3, it is a bit hard to tell how many levels of feature entropy each number of objects has. For example, the 10 objects condition has two small bends that make it look like there are only three levels. For visualization purposes, I would recommend adding markers to the plot. Additionally, the values on the x-axis don't seem to align with the actual levels of feature entropy that were used. I understand that different number of objects had different levels, making it impossible to accomodate all possible levels, but if at least some of them could be aligned that would make the plot easier to read.
- Also related to the scene description task, an explanation should be added of how the "edit distance between the true description of the scene and the model's description", the measure of errors, was computed. Does this mean that, for each (ground-truth) object in the scene, the most similar object (in terms of shape and color) in the description was chosen, and differences between that object and the ground truth were counted (e.g. color different = 1 error, shape different = 1 error)? How was, for example, a missing object in the description counted? As 2 errors, as both the color and shape were missing? A detailed explanation should be added.
- The visual analogy task is similarly missing a detailed explanation of the pair generation process. In particular, what were the relations included in the task? Figure 4 apparently shows a "sameness" relation, in which target pair 2 is more similar to the source pair as they both feature objects that differ in color. Was this the only relation included? Also, on page 8, line 272 you write that the incorrect target "shared only one of the relations" with the source pair. I assume that this means that in figure 4, for example, TP2 shares both the "same shape" and the "different color" relation with the source pair, while TP1 only shares "same shape". Making this explicit and clarifying with an example would make this part easier for the reader.
- The decomposed condition was only applied to the analogy task: was there a reason for this? As it's proposed as a general recipe for alleviating feature binding problems in VLMs, it would be more convincing if it was shown to work on a variety of tasks. The other tasks in the paper, in particular, seem to me like they would be amenable to be presented sequentially to the models. For example, a sequence of images containing only a few items at a time could be presented in visual search, simulating the way in which human participants serially scan such displays with eye movements in visual search experiments. If I am missing some practical reasons why conducting these experiments with a decomposed condition would be infeasible, or uninformative, I would be grateful to the authors if they could explain them.

**Questions:**

No particular questions. I have mentioned a few points that I believe should be clarified in the "weaknesses" section, as I believe the paper would benefit from the addition of their answers.

**Limitations:**

One limitation which has not been acknowledged is the use of closed-source models. This could compromise the reproducibility of this work, and will limit the ability of follow-up works to investigate the mechanisms underlying these findings.

---

> ### Author Rebuttal · Authors · 2024-08-07
>
> Thank you for your detailed review and valuable feedback. We appreciate the opportunity to address your comments and clarify our work. Below is our point-by-point response:
>
> **Concerns about response parsing**
>
> We thank the reviewer for pointing out this important concern. We have now eliminated the intermediate parsing step and ask the models to directly provide its response as output. We find that all of the same binding errors are present without secondary response parsing (see global rebuttal).
>
> **Explanation of metrics used in Scene Description task**
>
> We have updated our scene description task to explicitly evaluate the relationship between feature interference across objects and model performance. Instead of using entropy, we now estimate the "likelihood of a swap error," defined as the number of triplet pairs in the color/shape feature matrix where swapping two of the triplet features results in a binding error. We find that this new metric is significantly correlated with task performance, and provides a more elegant account of model behavior. See section A and B of the attached PDF for the updated results. Our dataset is generated by balancing the number of trials across each object count and feature diversity condition. Detailed descriptions of how this novel metric is computed and how our scene description dataset was generated are provided in the updated manuscript.
>
> Edit distance is defined as the number of objects the model failed to identify from the ground truth set of objects plus the number of objects erroneously identified (conjunction of shape/color for identified object not present in the ground truth). We have added a detailed description of the updated prompting strategy and a definition of edit distance in the revised manuscript.
>
> **Improved plotting for scene description task:**
>
> We appreciate the suggestion. We have implemented these changes in the updated plots to improve clarity (see section A of the attached PDF).
>
> **Explanation of RMTS task:**
>
> We have clarified that the relations are ‘same shape, different color’. The correct answer is Target Pair 2 because it satisfies both relations, while Target Pair 1 only satisfies one. We did not observe any evidence that the task was ambiguous for the VLMs. This clarification has been added to the revised manuscript.
>
> **Feasibility of running decomposed trial conditions on tasks other than RMTS**
>
> For counting and scene description, we believe that the decomposed condition would likely improve performance, but we did not test this because we felt that it may be viewed as trivial. For example, in the counting task, this would just amount to counting the number of images that are presented. For the search task, the number of objects exceeds the context window size for images (10 images max for GPT-4V).
>
> **Limitations of Closed Source Models**
>
> We acknowledge this limitation and have added experiments with LLaVA 1.5. While LLaVA did not respond consistently with our task instructions using the original prompts, modified prompts show that the subitizing limit in the counting task is preserved (see section C in attached PDF). However, the model struggles with search tasks (section F) and visual analogy. The absence of effects in LLaVA may be due to performance differences or prompting strategy.
>
> We hope these revisions and clarifications address your concerns. Thank you again for your insightful feedback.

---

> ### Comment · Reviewer_Ga5j · 2024-08-13
>
> I thank the authors for their rebuttal. My concerns were quite minor in the first place, and mostly related to presentation. Beyond responding to my own concerns, given that the authors have substantially expanded the paper with more thorough analyses, I have increased my score by one point.

---

### Official Review · Reviewer_MYwe · 2024-07-08

**Soundness:** 2
**Presentation:** 3
**Contribution:** 3
**Rating:** 7
**Confidence:** 4

**Summary:**

This work investigates the performance of the vision language model (VLM) GPT-4v and text-to-image model DALL-E-3 on multi-object reasoning tasks. According to the literature, humans perform well on multi-object tasks because they learn a compositional representation of the world and use serial processing to dynamically combine object features, overcoming the binding problem. However, in time-constrained tasks, they fail to use serial processing and rely on parallel processing, leading to illusory conjunction. In this work, the authors hypothesize, without testing, that VLMs learn a compositional representation for the sake of generalization but are not compelled to develop a serial-processing mechanism. According to the authors, this leads to failure in multi-object reasoning tasks, thus exhibiting the same limitations as humans in time-constrained tasks. The authors test these limitations on four multi-object reasoning tasks: counting, visual search, scene description, and visual analogy. In these tasks, the authors show that the performance of VLMs decreases as the number of objects increases, supporting the conclusion that VLMs are not able to solve the binding problem because they lack the mechanism to perform serial processing.

**Strengths:**

- The paper is straightforward and easy to understand. The authors did a good job making it accessible, and the presentation and sectioning help in grasping the content.
- This work seeks to understand the limitations of VLMs through the lens of human cognition, which could provide valuable insights.
- Originality in the problem statement: In terms of performance, VLMs in their normal regime are equivalent to humans in time-limited tasks.
- The investigation utilizes various types of experiments, each guided by studies conducted on humans, and they look easy to redo and the empirical results seem relevant.

**Weaknesses:**

My score would increase if the following weaknesses were addressed:
- The hypothesis that VLMs learn a compositional representation for generalization purposes could benefit from a reference if possible, since the discussion depends on that hypothesis.
- Despite conducting various types of experiments (four in total), each individual experiment lacks sufficient variability. For instance, in the counting experiment, although they vary the colors of the circles, they do not vary their sizes or shapes. Another example, for the poppout task it would have been nice to also alternate the color of target and distractors.
- The authors carried out the experiment on a very small group of language models (two).

Some typos:
- Figure 1, top right: the x-axis should be integers in the plot of GPT-4v counting performance
- Line 156, 188: should be ‘Figure 2’ and not ‘Figure 1’

**Questions:**

If we carefully engineer prompts for VLMs to sequentially focus and analyze each object in an image, considering they can learn a compositional representation as claimed, thus they should theoretically possess all the elements necessary to solve the binding problem? Can you elaborate on that?

**Limitations:**

See “Weaknesses”

---

> ### Author Rebuttal · Authors · 2024-08-07
>
> Thank you for your detailed review and valuable feedback. We appreciate the opportunity to address your comments and clarify our work. Below is our point-by-point response:
>
> **Concerns about generalizability and scope of our results**
>
> We thank the reviewer for suggesting to run additional control experiments and to analyze a broader range of models. We have run the following additional experiments to strengthen our work.
>
> 1. We have expanded our analysis to include five VLMs: GPT-4v, GPT-4o, Claude Sonnet 3.5, Google Gemini Ultra 1.5, and LLaVA 1.5. Additionally, we have expanded the text-to-image model experiments to include four models: DALL-E-3, Stable Diffusion Ultra, Google Parti, and Google Muse. The general effects are present across all models, with some minor deviations for LLaVA, which we suspect is due to its specific limitations relative to stronger open source models.
>
> 2. For the visual search task, we included a condition where the color of the target and distractors varies across each trial. The results show that model performance is comparable to the original experiment, with the same qualitative effect. Please refer to sections E and F of the attached PDF for the figure of results in an example model.
>
> 3. For the counting task, we introduced conditions where we manipulate the color and shape dimensions of the stimuli. Our findings support our hypothesis that capacity limits arise due to issues with representational interference. Performance is generally highest in the highest entropy condition (unique colors and shapes across all objects), lowest in the low entropy condition (uniform colors and shapes), and intermediate for the medium entropy conditions (either unique colors or shapes, but not both). Please refer to section C of the attached PDF for the figures of the updated results.
>
> **Evidence for compositional representations in VLMs/LLMs**
>
> The reviewer has requested that we cite evidence in support of the claim that VLMs learn compositional representations. We have included references to the following papers, which provide evidence that LLMs/VLMs form compositional representations (Lepori et al. 2023; Lewis et al. 2022; McCoy 2022; Yun et al. 2022, Yu et al. 2023), and agree that future work should directly investigate the representations themselves. We additionally note that if VLMs did not learn compositional representations, we would not expect to observe a different pattern of errors in the model behavior. The binding problem arises only when a system uses compositional representations, thus evidence for the binding problem can be viewed as indirect evidence for the presence of compositional representations.
>
> **Solving the binding problem with improved prompting?**
>
> It is difficult to say for sure, given the lack of transparency about closed-source model architectures, whether there might be some prompting approach that improves results. However, there are reasons to believe that prompting alone cannot resolve the binding problem in current models. First, we have already attempted to prompt the models to enumerate objects one at a time, which did not improve performance (indeed, the performance reported in both counting and search tasks is prompting the model to explicitly enumerate all objects). Second, there are architectural limitations:
>
> 1. VLM image encoders may struggle to maintain distinct object representations within a single embedding. Compressing multi-object scenes while preserving accurate feature bindings is challenging, potentially limiting the model's ability to process objects individually.
>
> 2. In autoregressive transformers, it is possible to perform some degree of serial processing through the output of the model (i.e. chain-of-thought), but in order to use this to sequentially attend to individual objects, there must be some clear way to index these objects in the model's output, which is often difficult for multi-object scenes.
>
> We have added an explanation of these factors to the revised manuscript and included relevant citations.
>
>
> We hope these revisions and clarifications address your concerns. Thank you again for your insightful feedback.
>
> **Citations**:
>
> Lepori, M., Serre, T., & Pavlick, E. (2023). Break it down: Evidence for structural compositionality in neural networks. *Advances in Neural Information Processing Systems*, 36, 42623-42660.
>
> Lewis, M., Nayak, N. V., Yu, P., Yu, Q., Merullo, J., Bach, S. H., & Pavlick, E. (2022). Does clip bind concepts? probing compositionality in large image models. *arXiv preprint* arXiv:2212.10537.
>
> McCoy, R. T. (2022). Implicit Compositional Structure in the Vector Representations of Artificial Neural Networks (Doctoral dissertation, Johns Hopkins University).
>
> Yun, T., Bhalla, U., Pavlick, E., & Sun, C. (2022). Do Vision-Language Pretrained Models Learn Composable Primitive Concepts?. *arXiv preprint* arXiv:2203.17271.
>
> Yu, D., Kaur, S., Gupta, A., Brown-Cohen, J., Goyal, A., & Arora, S. (2023). Skill-Mix: A flexible and expandable family of evaluations for AI models. *arXiv preprint* arXiv:2310.17567.

---

> > ### Comment · Reviewer_MYwe · 2024-08-10
> >
> > Thank you for thoroughly addressing all my questions and concerns in your rebuttal and revised paper. I appreciate the effort and clarity you’ve provided, and as a result, I have decided to increase my score by one point.

---

### Official Review · Reviewer_gAnB · 2024-07-12

**Soundness:** 3
**Presentation:** 3
**Contribution:** 2
**Rating:** 7
**Confidence:** 3

**Summary:**

**Summary:**
	The paper investigates the counting abilities of VLMs. It puts forward a cognitive science inspired perspective, where VLMs are fundamentally constrained in their processing of multi-object scenes due to a lack of serial processing mechanisms.

**Decision:**
	The paper is well structured and investigates an interesting (albeit a bit constrained) question concerning VLMs object reasoning abilities. While some of the conclusions seem overstated to me, I nonetheless recommend accept, since the authors present novel findings that improve the research community's understanding of VLMs visual reasoning abilities.

**Strengths:**

*Originality*
		The underlying question is timely and well motivated. The cognitive science derived approach is principled and the authors perform a number of interesting experiments to further delineate the issues underlying counting in VLMs.

*Quality*
		The submission is technically sound. I don't agree with all resulting claims, but the methods are appropriate.

*Clarity*
		The submission is clearly written and follows a good outline.

*Significance*
		The paper yields new data which allows for unique conclusions, however the scope of the investigation is somewhat limited.

**Weaknesses:**

I have brought it up in the questions sections.

**Questions:**

- Figure 1: There is a spike at 9 objects for both models. Do you have an idea on why this might be the case? [1] (which you already cite) also find the same spike and offer some potential explanations for this (see Figure 5).
- Answer parsing is again conducted by a large language model. I am wondering how accurate this parsing is. Did you compute any metrics to check the accuracy of the final parsed output or run some qualitative checks?
-  In line 189 you write "GPT-4v demonstrates human-like capacity constraints in its ability to perform visual search in multi-object settings." I was wondering what human level performance is in this task? It would be nice to have some more information to compare the model performance to. The same is true for line 127, where I'm not sure this is a fair comparison. Sure, the number of objects the models can count well are somewhat similar to the subitizing limit of humans, but this task should not require rapid estimation of the number of objects and therefore we would expect humans to be able to count well above 6 objects here anyways.
- The sentence in line 303 reads a bit odd, it should probably read "facilitate generalization" instead of facilitation and there might be a comma missing before?
- If I understand correctly you describe a different measure for the model performance for the last task in Section 4.2, would it not be possible to again report the proportion of correct answers here? I find it hard to compare this experiment with the previous experiments.
- To me the take aways seem overstated. What does lack of serial processing for VLMs mean? I don't think VLMs are fundamentally unable to sequentially process an image and therefore they should in theory be able to "selectively attend to individual objects one at a time". This is also why I don't completely follow the conclusion that the VLMs show "human-like capacity constraints". I would argue they should be able to do sequential processing and should therefore not be measured against the human subitizing limit anyways. Maybe you can offer some more details on this argument.

[1] Rane, S., Ku, A., Baldridge, J., Tenney, I., Griffiths, T., & Kim, B. (2024). Can Generative Multimodal Models Count to Ten? Proceedings of the Annual Meeting of the Cognitive Science Society, 46. Retrieved from https://escholarship.org/uc/item/8kz5787g

**Limitations:**

The paper is well structured and investigates an interesting (albeit a bit constrained) question concerning VLMs object reasoning abilities. While some of the conclusions seem overstated to me, I nonetheless recommend accept, since the authors present novel findings that improve the research community's understanding of VLMs visual reasoning abilities.

---

> ### Author Rebuttal · Authors · 2024-08-07
>
> Thank you for your detailed review and valuable feedback. We appreciate the opportunity to address your comments and clarify our work. Below is our point-by-point response:
>
> **Clarification on human-like capacity limits and the logic of our experiments**
>
> The reviewer has asked for clarification regarding human performance in the visual search task, and raised the concern that humans are indeed able to count beyond the subitizing limit when given enough time. Before directly addressing these issues, let us first briefly explain the types of capacity limits that we are interested in, and the conditions under which they arise in human cognition. It is widely acknowledged in cognitive science that there is a distinction between parallel (automatic) and serial (controlled, deliberative) modes of processing (Posner et al. 1980; Schneider et al. 1977; Treisman & Gelade 1980). The parallel mode of processing is dominant when people make rapid judgments, and displays classic capacity limits, such as difficulty counting beyond the subitizing limit, and difficulty in conjunctive search tasks. The serial mode of processing requires temporally extended processing, and is able to overcome these capacity limits, enabling precise counting for large numbers of objects, and serial search to identify specific feature conjunctions.
>
> Our hypothesis in this work is that the current generation of VLMs is primarily limited to the parallel mode of processing. We explain below some reasons for thinking that this might be the case (based on current understanding of the architectures and algorithms used), but, given the close-sourced nature of most current VLMs, our approach in this work is to test whether VLMs display behavioral signatures that resemble rapid parallel processing in human cognition. We find that VLMs do indeed display several specific phenomena that are diagnostic of parallel processing, including 1) difficulty counting beyond the subitizing limit, 2) greater difficulty in counting arrays of objects with uniform features than arrays with variable features (see new results in global rebuttal), 3) performance on disjunctive search tasks that is invariant to the number of objects, combined with decreased performance on conjunctive search tasks as a function of the number of objects, and 4) binding errors in scene description. Taken together, these phenomena provide indirect, but nevertheless highly specific, evidence that VLMs are limited to parallel processing. The fact that humans can complement this form of processing with serial processing – as in counting and serial search – is not at odds with our conclusions. Rather, it suggests a way that we can imagine augmenting VLMs to bring them closer to human performance.
>
> **Potential reasons for limited serial processing in current VLMs**
>
> We do not claim that serial processing is beyond the ability of any type of vision-language model, including future improvements of these systems, but there are at least two major reasons to suspect that the current generation of VLMs may suffer from difficulty with serial processing:
>
> 1. VLM image encoders may struggle to maintain distinct object representations within a single embedding. Compressing multi-object scenes while preserving accurate feature bindings is challenging, potentially limiting the model's ability to process objects individually.
> 2. In autoregressive transformers, it is possible to perform some degree of serial processing through the output of the model (i.e. chain-of-thought), but in order to use this to sequentially attend to individual objects, there must be some clear way to index these objects in the model's output, which is often difficult for multi-object scenes.
>
> These are certainly not fundamental limitations. It is easy to imagine that serial processing can be improved in VLMs through the use of object-centric visual representations, or the use of recurrent rather than feedforward transformers, and there is indeed some recent work in this direction (You et al. 2023; Webb et al. 2024). Our results suggest that VLMs will benefit from the incorporation of these stronger inductive biases for serial processing. We have added further discussion of these issues to the paper.
>
> **Possibility of errors in LLM answer parsing**
>
> This is an excellent point. To avoid concerns over response parsing, we have updated our prompts so that VLMs directly return their responses in a form that we can evaluate without a secondary parsing model. We find that all of the same binding errors are present without secondary response parsing (see global rebuttal).
>
> **Spike in counting performance for 9 objects**
>
> Thanks for highlighting this. We've run extended analysis across more models (see global rebuttal) and don't universally observe this trend. We agree it's interesting – like Rane et al. 2024, we hypothesize it may result from 3x3 grid arrangements in training data.
>
>
> **Citations:**
>
> Posner, M. I., Snyder, C. R., & Davidson, B. J. (1980). Attention and the detection of signals. *Journal of experimental psychology: General*, 109(2), 160.
>
> Rane, S., Ku, A., Baldridge, J., Tenney, I., Griffiths, T., & Kim, B. (2024). *Can Generative Multimodal Models Count to Ten?*. In Proceedings of the Annual Meeting of the Cognitive Science Society (Vol. 46).
>
> Schneider, W., & Shiffrin, R. M. (1977). Controlled and automatic human information processing: I. Detection, search, and attention. *Psychological review*, 84(1), 1.
>
> Treisman, A. M., & Gelade, G. (1980). A feature-integration theory of attention. *Cognitive psychology*, 12(1), 97-136.
>
> Webb, T., Mondal, S. S., & Cohen, J. D. (2024). Systematic visual reasoning through object-centric relational abstraction. *Advances in Neural Information Processing Systems*, 36.
>
> You, Haoxuan, et al. "Ferret: Refer and ground anything anywhere at any granularity." *arXiv preprint* arXiv:2310.07704(2023).

---

> > ### Author Response · Authors · 2024-08-07
> >
> > ## Accuracy measure for scene task
> > When submitting our rebuttal, we forgot to include our response to the question about including an accuracy measure for the scene description task. In this task, we chose to focus on 'edit distance' because it is a more sensitive measure than accuracy for this task. The models very often make at least one mistake for each image, meaning that accuracy (proportion of trials for which the entire scene description is correct) is very low, and thus not as sensitive for testing our specific hypothesis about binding errors. However, we agree that it is useful to include an accuracy measure for the purpose of comparing across tasks. We have now included these plots in the appendix. Thank you for the suggestion.

---

> > > ### Comment · Reviewer_gAnB · 2024-08-12
> > >
> > > I thank the authors for their detailed explanation on serial and parallel processing. I now have a better understanding of the assumptions and objectives behind the submission. Since, as far as I can see, all of my major concerns were addressed, I will increase my score.

---

### Official Review · Reviewer_N3JA · 2024-07-14

**Soundness:** 3
**Presentation:** 4
**Contribution:** 3
**Rating:** 7
**Confidence:** 4

**Summary:**

The paper puts forth the hypothesis that VLMs suffer from similar constraints as humans in multi-object visual processing, especially
with respect to "binding".  Results from cognitive science show human constraints in numerical estimation, visual search, and relation
identification when not able to do serial processing, which is required for "binding" different object features together. The paper hypothesizes that lack of serial processing and binding in VLMs is responsible for failures in these tasks.  The paper gives experimental
results that support this hypothesis.

**Strengths:**

This is a good paper -- it gives a cognitive-science-based approach to understanding VLM behavior. It is well-written and has a mostly clear discussion of hypotheses, experiments, and results.

**Weaknesses:**

The paper leaves out some important information.  For each task, the authors prompted GPT-4v to describe the input images, but as far as I can tell, they did not give any evaluation of how well GPT-4v was able to describe them.  Other work (e.g., http://www.arxiv.org/abs/2407.06581) has shown that GPT-4v has trouble describing such synthetic images and often will hallucinate.

Here, the authors should include the results on how well GPT-4v can even parse what is in the images (including shapes, colors, etc.) and the authors should assess whether that failure to describe simple images is related to its failure on these tasks.  To what extent are the failures a matter of lack of binding features, or of missing or hallucinating features?

A couple of typos/corrections:

In Section 3, "Figure 1" should be "Figure 2".

In the caption for Figure 2, "blue column" should be "red column" and "red column" should be "blue column".

**Questions:**

Figure 2 caption: "Performance for 2D and 3D task variants" -- performance of what?  GPT-4v?  I also suggest in that caption that you
should say what the targets are for each search (red circle, red "L", red sphere, red sphere).

In Section 3.1., explain what the task is for GPT-4V: "Is there a red circle?" "Is there a red L?  This is explained in the appendix (A.2) but should be explained here in the main paper.

Looking at the visual search task as described in A.2, it seems that the task wasn't actually "search" in the sense that, unlike humans, GPT-4V doesn't have to localize the target object.  In these kinds of tasks, aren't humans asked to localize the target?  How does this affect the comparability of the results between GPT-4V and humans on this task?

In Section 3.1, explain why in half the images, targets were left out. This doesn't make sense unless readers know what the specific task is.

In Section 3.1 you say "VLMs demonstrate similar capacity constraints to humans."  What do you mean by "similar"?  Do humans show a similar success/failure curve as those in Figure 2? Or is the similarity just that humans perform well on disjunctive search and badly on
conjunctive search (if not enough time for serial processing)?

In Section 3.2, explain how the "accuracy" of the model is calculated.

In Section 4.2 you refer to "the true description of the scene". Where did this "true description" come from?  Also, define "edit
distance".  I was also confused because each scene could have different correct descriptions with different levels of detail.  Was it possible that the VLM gave one version of a correct description but it was counted as wrong because it included details that the "true description" didn't contain?

I was also confused by the visual analogy task in Figure 4.  It seems to me that both Target Pair 1 and Target Pair 2 could be considered
the correct answer.  The source pair has two obvious relations: "same star shape" and "different color".  The former would map to Target 1 and the latter would map to Target 2.  Which Target did you consider correct, and why?

**Limitations:**

The authors do a good job of discussing limitations.

---

> ### Author Rebuttal · Authors · 2024-08-07
>
> Thank you for your detailed review and valuable feedback. We appreciate the opportunity to address your comments and clarify our work. Below is our point-by-point response:
>
>
> **GPT-4v’s ability to describe synthetic images, and relationship to concurrent work (‘Vision language models are blind’)**
>
> Thank you for highlighting this highly relevant concurrent work, which we have now cited in the revised manuscript. We completely agree that VLMs struggle to accurately describe multi-object scenes. In this work, we propose an explanation for why VLMs show this difficulty. Specifically, we propose that it results primarily from an inability to accurately bind features at the object level. This is distinct from explanations that involve hallucinating features that aren’t present in the image (see additional results in section below for an analysis of these failure modes).
>
>
> **Analysis of binding failures vs. hallucinations**
>
> We agree that it is important to distinguish binding failures from the hallucination of individual features that aren’t actually present in the image. We have now included an analysis to account for this. We found that  the model rarely (<25% of errors) generates descriptions containing features not present in the trial. When this does happen, it is typically due to the misidentification of a feature with a close neighbor (e.g., attributing the color cyan to a blue shape). Most errors in this task can be attributed to binding errors where the model incorrectly attributes a feature to the wrong object.
>
>
> **Clarifications regarding visual search task**
>
> In these tasks, subjects are typically asked to identify whether the target is present, so the task is comparable. It is called ‘search’ in the literature, but we agree that it doesn’t necessarily involve localization. We have clarified this in the revised manuscript. We have also clarified why half of the images do not involve targets (because it is really a detection task, there must be some trials without targets in order to obtain a meaningful accuracy estimate).
>
> Regarding the similarity between human and VLM capacity constraints, it is difficult to perform a direct comparison because the original tasks involve a response time measure, and there is no clear analog of response time in VLMs. The classic finding is that response time increases as a function of the number of objects in the conjunctive search task, but is unaffected by the number of objects in the disjunctive search task. Because we do not have a response time measure for VLMs, we instead evaluate accuracy. We find a similar qualitative pattern (accuracy varies as a function of the number of objects in the conjunctive search task, but is unaffected in the disjunctive search task).
>
> We have also added an explanation of how accuracy is calculated in this task. Accuracy is computed as the proportion of trials that are correct. The following accuracy equation has been included in the supplemental for clarity: $acc = \frac{nCorrect_{target} + nCorrect_{distractors}}{nTrials}$
>
>
> **Clarification regarding ‘true description’ and ‘edit distance’ in the scene description task**
>
> To address these concerns, in the most recent version of the paper, we have updated the prompts so that the model directly returns a json of the trial objects’ shapes and colors (rather than an open-ended description of the scene). Edit distance here is defined as the number of objects that the model failed to identify from the ground truth json plus number of objects that the model erroneously identified (conjunction of shape/color not present in the ground truth json). We have added a description of the updated prompting strategy and a definition of edit distance in the revised manuscript.
>
>
> **Clarification regarding correct answer in example visual analogy problem**
>
> We have clarified that the correct target pair must share both relations (shape and color) with the source pair. The correct answer in the example is Target Pair 2 because it satisfies both the ‘same shape’ and ‘different color’ relations. Target Pair 1 only satisfies one relation. We did not observe any evidence that the task was ambiguous for the VLMs. This clarification has been added to the revised manuscript.
>
> We have also corrected typos identified, and implemented other clarifications (in the caption for figure 2, and the description of the visual search task).
>
> We hope these revisions and clarifications address your concerns. Thank you again for your insightful feedback.

---

> > ### Comment · Reviewer_N3JA · 2024-08-09
> > **Thanks for the rebuttal**
> >
> > The authors' rebuttal and paper revisions respond to all the concerns I had.  Thank you to the authors for the thorough rebuttal.

---

### Author Rebuttal · Authors · 2024-08-07

Dear Reviewers,

We sincerely thank you for your thoughtful and constructive feedback. In response to your comments, we have conducted several new analyses that we believe have significantly strengthened our work:

1. **Extended Model Analysis**: We expanded our study to include five VLMs (GPT-4v, GPT-4o, Claude Sonnet 3.5, Google Gemini Ultra 1.5, and LLaVA 1.5) and four text-to-image models (DALL-E-3, Stable Diffusion Ultra, Google Parti, and Google Muse). This broader analysis demonstrates the generalizability of our findings across different model architectures.
2. **Additional Counting Experiments**: We introduced new conditions manipulating color and shape dimensions of stimuli. We explored four conditions with varying levels of feature entropy: a low-entropy condition where both color and shape were uniform across all stimuli; two medium-entropy conditions where either color or shape were uniform while the other varied uniquely across stimuli; and a high-entropy condition where both color and shape were unique for each stimulus. Results support our hypothesis that capacity limits arise due to issues managing representational interference across objects, with performance highest in high-entropy conditions and lowest in low-entropy conditions, with graded levels of performance in between.
3. **Refined Scene Description Task**: We developed a novel metric, 'number of triplets', which is directly predicted by our theory of object representation in neural networks. This metric quantifies the potential for binding errors in multi-object scenes by counting feature triplets. A triplet consists of three objects where:  two objects share one feature (such as shape), two objects share a different feature (such as color), and critically, one object possesses both of these shared features. Our theory predicts that these triplets create opportunities for feature misattribution, potentially leading to incorrect object descriptions. For example, a scene containing a red circle, red square, and blue square could erroneously yield a description including a non-existent blue circle. The number of triplets in a scene directly corresponds to the likelihood of binding errors: more triplets create more opportunities for feature swaps, increasing error potential. This metric, grounded in our theoretical framework, provides a direct evaluation of model performance in maintaining correct feature bindings. We find across all models evaluated that performance is strongly predicted by this metric (see sections A and B in attached PDF), further confirming that binding errors are one of the primary limitations in these models.
4. **Additional Control Experiments**: For the visual search task, we included a condition where target and distractor colors vary across trials (last subplot in section E of attached PDF). Results show comparable performance to the original experiment, reinforcing the robustness of our findings.
5. **Clarified Methodology**: We've updated our prompting strategies to remove any post-hoc response parsing with a secondary model to address the reviewer concerns.
6. **Analysis of Binding Failures vs. Hallucinations**: We found that <25% of errors involve hallucinating features not present in the image, with most errors attributable to binding failures.

We provide detailed responses to the concerns and questions identified by each reviewer in the individual responses below.

---

### Decision · Program_Chairs · 2024-09-25

**Decision:**

Accept (poster)

**Comment:**

The paper takes a cognitive science perspective to investigate the ability of modern VLMs on multi-object reasoning tasks. The key result is that models perform similar to humans when they are processing is forced to be time-constrained, ie they have to rely on parallel as opposed to serial processing. The implication is that models limitations stem from an inability to leverage serial as opposed to parallel processing. The reviewers unanimously found the paper clear, insightful and all experiments to be carried out well. The AC thus recommends the paper to be accepted for rigorously testing limitations of current VLMs.